# SERINC5 restricts influenza virus infectivity

Fei Zhao[1][⊙], Fengwen Xu[1][⊙], Xiaoman Liu[1][⊙], Yamei Hu[1], Liang Wei[1], Zhangling Fan[1], Liming Wang[1], Yu Huang[1], Shan Mei[1], Li Guo[2], Long Yang[3], Shan Cen[4], Jianwei Wang[2]*, Chen Liang[5]*, Fei Guo[1]*

**1** NHC Key Laboratory of Systems Biology of Pathogens, Institute of Pathogen Biology, and Center for AIDS Research, Chinese Academy of Medical Sciences & Peking Union Medical College, Beijing, P. R. China, **2** NHC Key Laboratory of Systems Biology of Pathogens and Christophe Mérieux Laboratory, Institute of Pathogen Biology, Chinese Academy of Medical Sciences & Peking Union Medical College, Beijing, P. R. China, **3** School of Integrative Medicine, Tianjin University of Traditional Chinese Medicine, Tianjin, China, **4** Institute of Medicinal Biotechnology, Chinese Academy of Medical Sciences & Peking Union Medical College, Beijing, P. R. China, **5** McGill University AIDS Centre, Lady Davis Institute, Jewish General Hospital, Montreal, Quebec, Canada

⊙ These authors contributed equally to this work.
* wangjw28@163.com (JW); chen.liang@mcgill.ca (CL); guofei@ipb.pumc.edu.cn (FG)

**Data Availability Statement:** All relevant data are within the manuscript and its Supporting information files.

**Funding:** This study was supported by funds from CAMS Innovation Fund for Medical Sciences (CIFMS 2021-1-I2M-038 and 2018-I2M-3-004) to

## Abstract

SERINC5 is a multi-span transmembrane protein that is incorporated into HIV-1 particles in producing cells and inhibits HIV-1 entry. Multiple retroviruses like HIV-1, equine infectious anemia virus and murine leukemia virus are subject to SERINC5 inhibition, while HIV-1 pseudotyped with envelope glycoproteins of vesicular stomatitis virus and Ebola virus are resistant to SERINC5. The antiviral spectrum and the underlying mechanisms of SERINC5 restriction are not completely understood. Here we show that SERINC5 inhibits influenza A virus infection by targeting virus-cell membrane fusion at an early step of infection. Further results show that different influenza hemagglutinin (HA) subtypes exhibit diverse sensitivities to SERINC5 restriction. Analysis of the amino acid sequences of influenza HA1 strains indicates that HA glycosylation sites correlate with the sensitivity of influenza HA to SERINC5, and the inhibitory effect of SERINC5 was lost when certain HA glycosylation sites were mutated. Our study not only expands the antiviral spectrum of SERINC5, but also reveals the role of viral envelope glycosylation in resisting SERINC5 restriction.

## Author summary

SERINC5, a multi-span transmembrane protein, is incorporated into HIV-1 particles in producing cells and inhibits HIV-1 entry. SERINC5 impairs the infectivity of HIV-1 particles by disrupting viral glycoprotein clusters and hindering viral membrane fusion. Beside HIV-1, pseudo viruses with envelope proteins from feline endogenous retrovirus (RD114), influenza A virus (H7), rabies virus and lymphocytic choriomeningitis virus was shown to be restricted by SERINC5. Here, we demonstrate SERINC5 restriction of influenza A virus and also reveal the heterogeneity of influenza A virus susceptibility to SERINC5. Hence our data attribute the restriction function of SERINC5 to influenza A virus

F.G., from the National Key Plan for Scientific Research and Development of China (2018YFE0107600) to F.G., from the Ministry of Science and Technology of China (2018ZX10301408-003) to F.G., from the National Natural Science Foundation of China (82072288) to F.G., from the Canadian Institutes of Health Research (CCI-132561) to C.L. The funders had no role in study design, data collection and analysis, decision to publish, or preparation of the manuscript.

**Competing interests:** The authors have declared that no competing interests exist.

in both virus producer and virus target cells. We also propose the mechanism of escaping SERINC5 restriction by altering HA glycosylation.

## Introduction

Serine incorporator (SERINC) proteins participate in the synthesis of serine-containing lipids including sphingomyelin and phosphatidylserine [1]. SERINC proteins are highly conserved in eukaryotes from yeast (encode one *SERINC* gene) to humans (encode five *SERINC* genes). SERINC proteins are integral membrane proteins with 9 to 11 membrane-spanning regions [1]. Among the five human SERINC proteins, SERINC3 and SERINC5 have been reported to inhibit HIV-1 infectivity, and this activity is antagonized by HIV-1 Nef protein [2–4]. SERINC5 is localized to the plasma membrane and gets incorporated into HIV-1 particles [5]. SERINC5 impairs the infectivity of HIV-1 particles by disrupting viral glycoprotein clusters and hindering viral membrane fusion [6]. Incorporation of SERINC5 into HIV-1 particles also increase HIV-1 susceptibility to the inhibition of neutralization antibodies, suggesting an effect of SERINC5 on the conformation of viral glycoprotein [7,8]. HIV-1 Nef, as well as GlycoGag of murine leukemia virus (MLV) and S2 protein of equine infectious anemia virus (EIAV), counteract SERINC5 by decreasing the expression of SERINC5 at the plasma membrane and excluding it from virions [9–12].

SERINC5 also inhibits viruses other than retroviruses, but not by impairing virus infectivity. A recent study reported that SERINC5 inhibits classical swine fever virus through interacting with RNA sensor protein MDA5 and enhancing MDA5-dependent interferon response [13]. Another study showed that SERINC5 strongly inhibits the secretion of hepatitis B virus particles [14]. It appears that SERINC5 can restrict a relatively wide spectrum of different viruses via various mechanisms. Indeed, here we report that SERINC5 potently inhibits influenza A virus (IAV) infection. Importantly, SERINC5 not only impairs the infectivity of IAV particles, but also inhibits the entry of IAV when expressed in virus target cells. WSN and PR8 strain were less sensitive to SERINC5 when hemagglutinin (HA) glycosylation sites were mutated. We conclude that site-specific glycosylation of influenza A virus HA determines SERINC5 restriction of influenza viruses.

## Result

### SERINC5 inhibits IAV HA-mediated infection

It has been reported that the envelope protein of some HIV-1 strains such as YU-2 is resistant to SERINC5 inhibition. In addition, vesicular stomatitis virus (VSV) glycoprotein and Ebola virus glycoprotein, when used to pseudotype HIV-1 particles, have also been shown to resist SERINC5 in producing cells. We asked whether SERINC5 inhibits the infection mediated by IAV envelope protein HA. We thus co-transfected HEK293T cells with HIV-1 DNA ΔEnv/NefG2A, which does not express Env and has the NefG2A mutation, and HA and neuraminidase (NA) DNA of the IAV strain WSN, in the presence of increasing levels of SERINC5. In agreement with previous report, SERINC5 decreased the infectivity of virus particles carrying HIV-1 HXB2 envelope protein, but did not affect the infectivity of VSV-G or YU-2 Env pseudotyped viruses (Fig 1A). Viruses carrying WSN HA and NA were inhibited by SERINC5 in a dose-dependent manner (Fig 1A). Results of p24 ELISA and Western blots showed that SERINC5 did not affect the amounts of viral particles produced in the supernatants, nor viral protein expression in cells (Fig 1B and 1C). To determine whether HIV-1 Nef can counter

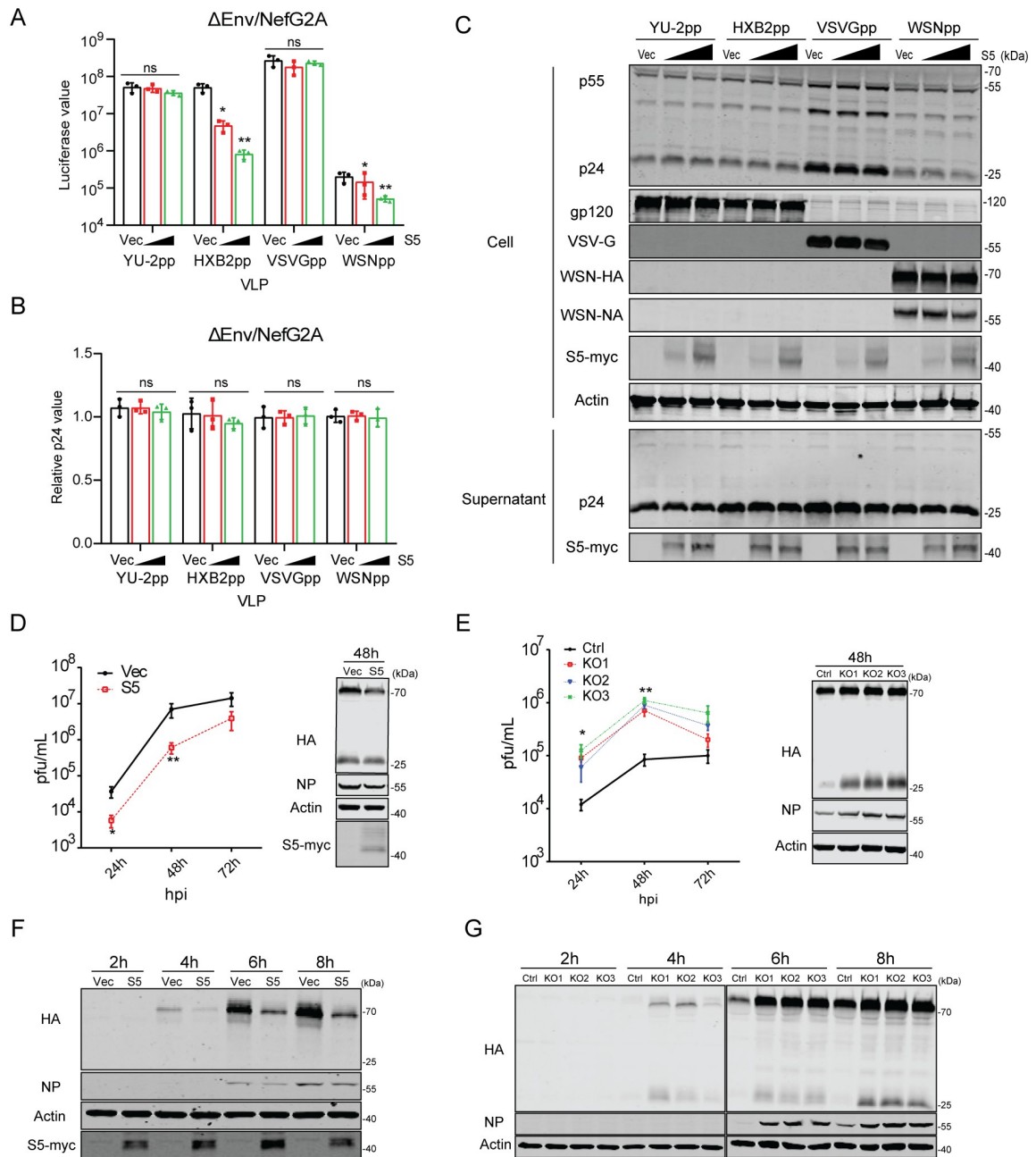

**Fig 1. Exogenous SERINC5 inhibits IAV infection.** A-C: Pesudoviruses were produced by transfecting HEK293T cells using plasmid DNA ΔEnv/NefG2A HIV-1[NL4-3] (containing mutated Nef) together with Env (HXB2), Env (YU-2), VSV-G, or HA/NA, and the indicated amounts of SERINC5 plasmid or empty vector. (A) Infectivity of the pseudoviruses was measured by infection of the TZM-bl indicator cells with equal p24[Gag] amounts of viruses. (B) Viral particle production was quantified by p24[Gag] ELISA. (C) Expression of SERINC5, and p24 in the cell lysates and supernatants were examined by Western blotting. HIV-1 pseudovirus particles were purified by ultracentrifugation through a 20% sucrose cushion. YU-2pp, pseudovirus with YU-2 Env. HXB2pp, pseudovirus with HXB2 Env. VSVGpp, pseudovirus with VSV-G protein. WSNpp, pseudovirus with WSN HA/NA proteins. D-E: Effects of exogenous SERINC5 on influenza A virus long-term infection. A549 cells, either stably overexpressing exogenous SERINC5 (MOI = 0.001) (D) or knocked out of SERINC5 (MOI = 0.0001) (E), were infected with virus A/WSN/33. Virus titers in the culture supernatants were determined by plaque assays. 48 h post infection, cells were harvested to measure the expression of SERINC5, HA and NP proteins. F-G: Effects of exogenous SERINC5 on influenza A virus short-term infection. SERINC5-overexpressing A549 cells (MOI = 5) and SERINC5-knockout cells (MOI = 1) were incubated with WSN. 2h, 4h, 6h and 8h post infection cells were collected and protein expression were determined as described above. Results shown are the averages of three independent experiments. Statistical significance was calculated with unpaired $t$-test. ns: not significant; *: $P<0.05$; **: $P<0.01$.

SERINC5 inhibition of WSN HA and NA-pseudoviruses, we used HIV-1 DNA ΔEnv/Nefwt which has the wild type Nef to produce pseudovirus particles. The results showed that Nef prevented SERINC5 from being incorporated into virus particles, and rescued the infection of HXB2 Env or WSN HA protein pseudotyped virus particles in the presence of SERINC5 (S1A–S1C Fig). The viral membrane is embedded with the M2 ion channel protein, and the hemagglutinin (HA) and neuraminidase (NA). Inside the virion, viral ribonucleoprotein (vRNP) complexes associates with the viral matrix protein M1 which associates with the inner leaflet of viral membrane [15]. M1 participates in viral particle assembly and can bind to the cell plasma membrane and/or the cytoplasmic tail of viral transmembrane proteins [16]. When these IAV proteins were co-expressed during the production of pseudovirus particles, viral infectivity increased, in agreement with what was previously reported [17–19]. However, SERINC5 still exhibited inhibition regardless of the presence of IAV proteins NA, M1 or M2 (S1D and S1E Fig), indicating that none of these IAV proteins is able to counter SERINC5.

HIV-1 DNA ΔEnv/NefG2A used here expresses accessory proteins Vif, Vpu, and Vpr that can evade cellular antiviral defense [20–23]. To rule out any potential effect of these viral proteins on SERINC5 inhibition of IAV, we used the lentiviral vector system that only carry HIV-1 Gag/Pol proteins. Strong inhibition by SERINC5 was observed when HXB2 Env or HA/NA proteins were used to pseudotype virus particles (S2 Fig). Together, these data suggest that SERINC5 inhibits the infectivity of virus particles carrying IAV envelope proteins HA and NA and this inhibition is antagonized by HIV-1 Nef protein.

## SERINC5 inhibits IAV infection

Next, we studied to which extent SERINC5 inhibits IAV infection. First, we generated A549 cell line stably expressing SERINC5, and monitored WSN infection at 24, 48 and 72 hours. SERINC5, IAV NP and HA proteins were detected at 48 h post infection by Western blots. SERINC5 overexpression reduced IAV titers by 10-fold (Fig 1D), in agreement with the decrease of viral NP and HA proteins in the infected cells. We next used CRISPR-Cas9 to generate SERINC5-knockout A549 cell clones. DNA sequencing confirmed the genetic changes that led to knockout of SERINC5 in three independent cell clones (S3 Fig). When these SERINC5-knockout cells were exposed to WSN infection, a 10-fold increase in viral titer was detected in the supernatants (Fig 1E), which is supported by the increase of viral NP and HA proteins in the infected cells. We also examined SERINC5 inhibition of IAV in HEK293T cells. SERINC5 overexpression reduced IAV titers in the culture supernatants at 24 h.p.i by 10-fold (S4A and S4B Fig), and SERINC5 knockdown increased IAV infection in HEK293T cells (S4C and S4D Fig).

There are five *SERINC* genes in the human genome. Of these five human SERINC proteins, all but SERINC2 can potentially restrict the infectivity of HIV-1. To determine whether other SERINC proteins inhibit IAV infection, we transfected HEK293T cells with each of the five *SERINC* genes cDNA, followed by infection with WSN. Results showed that only SERINC5 strongly reduced the production of infectious WSN IAV (S4E Fig).

We next performed WSN infection and monitored infection at 2, 4, 6 and 8 hours post infection. A decrease in viral NP and HA proteins was detected in SERINC5-overexpressing A549 cells (Fig 1F), as opposed to an increase in SERINC5-knockout cells (Fig 1G). Notably, the reduced expression of HA was observed as early as 4 h.p.i, suggesting that SERINC5 likely restricts the early steps of IAV infection when expressed in virus targeting cells. To further test this possible mechanism of inhibition, we transiently transfected A549 cells with SERINC5-mCherry, followed by WSN infection. Results of confocal microscopy showed lower levels of viral HA protein in the SERINC5-mCherry transfected cells compared with the untransfected

neighbor cells (S4F Fig). We also detected viral NP protein in WSN infected, SERINC5-over-expressing cells at 4 h.p.i. Again, lower levels of viral NP was expressed in SERINC5-positive cells (S4G Fig).

To further determine that SERINC5 inhibits the early steps of WSN infection, we performed the single cycle infection assay using WSN HA and NA pseudotyped ΔEnv/Nef-G2A-GFP viruses, to infect SERINC5-overexpressing or SERINC5-knockout A549 cell lines. SERINC5 overexpression inhibits WNS HA/NA-mediated infection (Fig 2A), whereas SERINC5 knockout increased virus infection (Fig 2B). To exclude the possible effect by HIV-1 accessory proteins, the same experiment was performed with the lentiviral vector, and the same observation was made (Fig 2C–2F).

Lastly, we used the single-cycle WSN-Gluc virus to infect SERINC5-overexpressing or SERINC5-knockout A549 cell lines. The WSN-Gluc virus has the HA segment replaced to express secreted *Gaussia* luciferase but maintains the HA package sequence. With HA provided *in trans*, WSN-Gluc can complete one round of infection (Fig 2G) [24]. The results showed that SERINC5 overexpression in the virus target cells reduced virus infection by 5-fold, and consistently, SERINC5-knockout increased WSN-Gluc infection (Fig 2H and 2I). Together, these results demonstrate that SERINC5 is able to protect the target cells from IAV through restricting the early steps of IAV infection.

## SERINC5 inhibits IAV entry by preventing virus-cell membrane fusion

To determine which step of IAV infection is inhibited by SERINC5 in the target cells, we first measured the effect of SERINC5 on sialic acid expression on the cell surface. We used Alexa Fluor-conjugated *sambucus nigra agglutinin* (SNA) and Biotinylated *Maackia amurensis* lectin II (MAL-II), which specifically recognizes α-2, 6 SA and α-2, 3 SA, to quantify the level of cell surface sialic acid. The results of flow cytometry showed that neither overexpression nor knockdown of SERINC5 changed cell surface expression of sialic acid (S5A and S5C Fig). We then examined IAV WSN attachment to cell surface by detecting viral HA protein with confocal microscope, after incubation of A549 cell with WSN virus at 4˚C for 1 hour. No change of HA levels was observed when SERINC5 was either overexpressed or knocked out (S5B and S5D Fig), which suggests that SERINC5 does not affect the attachment of IAV particles to the cell surface.

Next, we monitored the delivery of IAV RNA into the cytoplasm by real-time RT-PCR, and observed an increase of IAV RNA in SERINC5-knockout A549 cells 0.5 hours after infection, and a decrease in SERINC5-overexpressing cells (Fig 3A and 3B). To validate this observation, we visualized IAV RNA entry into the cytosol by performing RNA FISH using a specific probe targeting viral PB1 RNA. Fewer IAV RNA was detected in SERINC5-overexpressing A549 cells (Fig 3C), more in SERINC5-knockout cells (S6A Fig). Delivery of IAV RNA into the cytoplasm requires the disassembly of the IAV capsid, which can be monitored by staining of M1 protein. This process can be quantified by visualizing the redistribution of M1 from particle punctate to diffused signals [25]. mCherry- or mCherry-S5-transfected A549 cells were synchronously infected with WSN and M1 localization was visualized at various times post infection. We found that more punctate M1 was observed in mCherry-S5 positive cells than in mCherry positive control cells at various times post infection (S6B Fig, white arrow). Both results demonstrate that SERINC5 prevents IAV disassemble and vRNA release.

Lastly, we performed virus fusion assay to determine the effect of SERINC5 on IAV HA/NA-mediated viral entry. We first used authentic influenza virus to examine virus fusion. Briefly, influenza A virus WSN were dually labeled with DiOC18 and R18 which were incorporated into the membrane of WSN particles. The green fluorescence is suppressed to a level

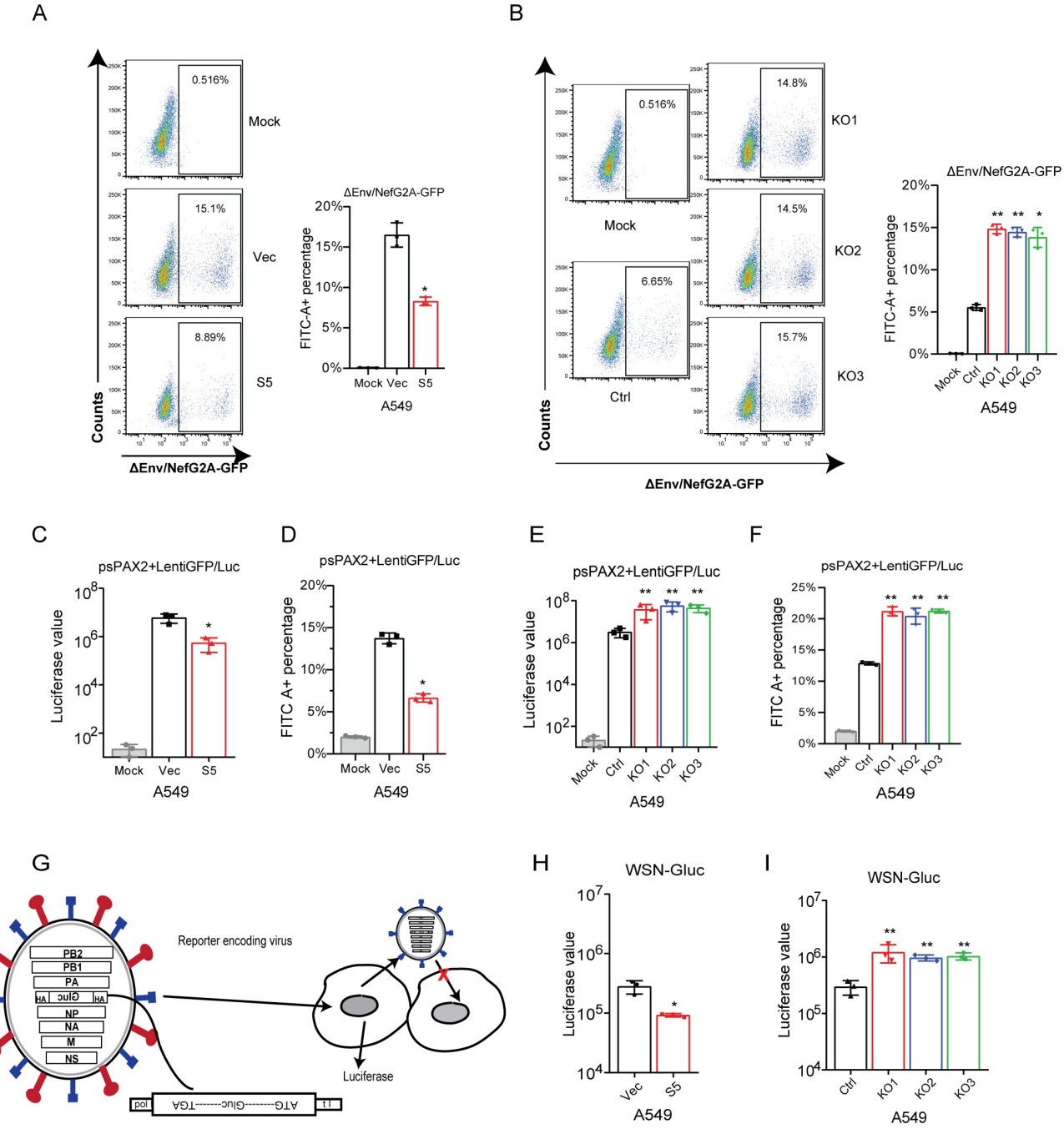

**Fig 2. SERINC5 inhibits IAV infection when expressed in virus target cells.** A-B: Pseudoviruses ΔEnv/NefG2A-GFP HIV-1[NL4-3] bearing IAV HA/NA proteins were used to infect A549 SERINC5-overexpressing cells (20ng p24 antigen) or SERINC5-knockout cells (10ng p24 antigen). Virus infection was examined by scoring GFP-positive cells with flow cytometry. C-F: Pseudovirions psPAX2 with influenza HA/NA proteins infected A549 SERINC5-overexpressing cell (20ng p24 antigen) (C, D) and SERINC5-knockout cell (10ng p24 antigen) (E, F). Half of the A549 cells were lysed to measure luciferase activity (C, E), the other half of A549 cells were fixed to measure pseudovirus infectivity by flow cytometry. Percentages of FITC-positive cells are shown in the column (D, F). G: Illustration of IAV for single-round infection. The HA ORF was replaced by Gluc. HA protein was expressed from the plasmid DNA. The expression plasmid of HA and 7 plasmids encoding IAV genomes were co-transfected to obtain the single-round IAV which can infect cells and produce secreted Gluc, but does not express HA protein and does not produce infectious virus particles for the next round of infection. This IAV is competent for a single-round infection. H-I: The single-round IAV was used to infected A549 SERINC5-overexpressing cells (MOI = 0.05) (H) or SERINC5-knockout cells (MOI = 0.01) (I). Levels of Gluc in the culture supernatants were measured 8 h.p.i. Results shown are the averages of three independent experiments. Statistical significance was calculated by unpaired *t*-test. ns: not significant; *: *P*<0.05; **: *P*<0.01.

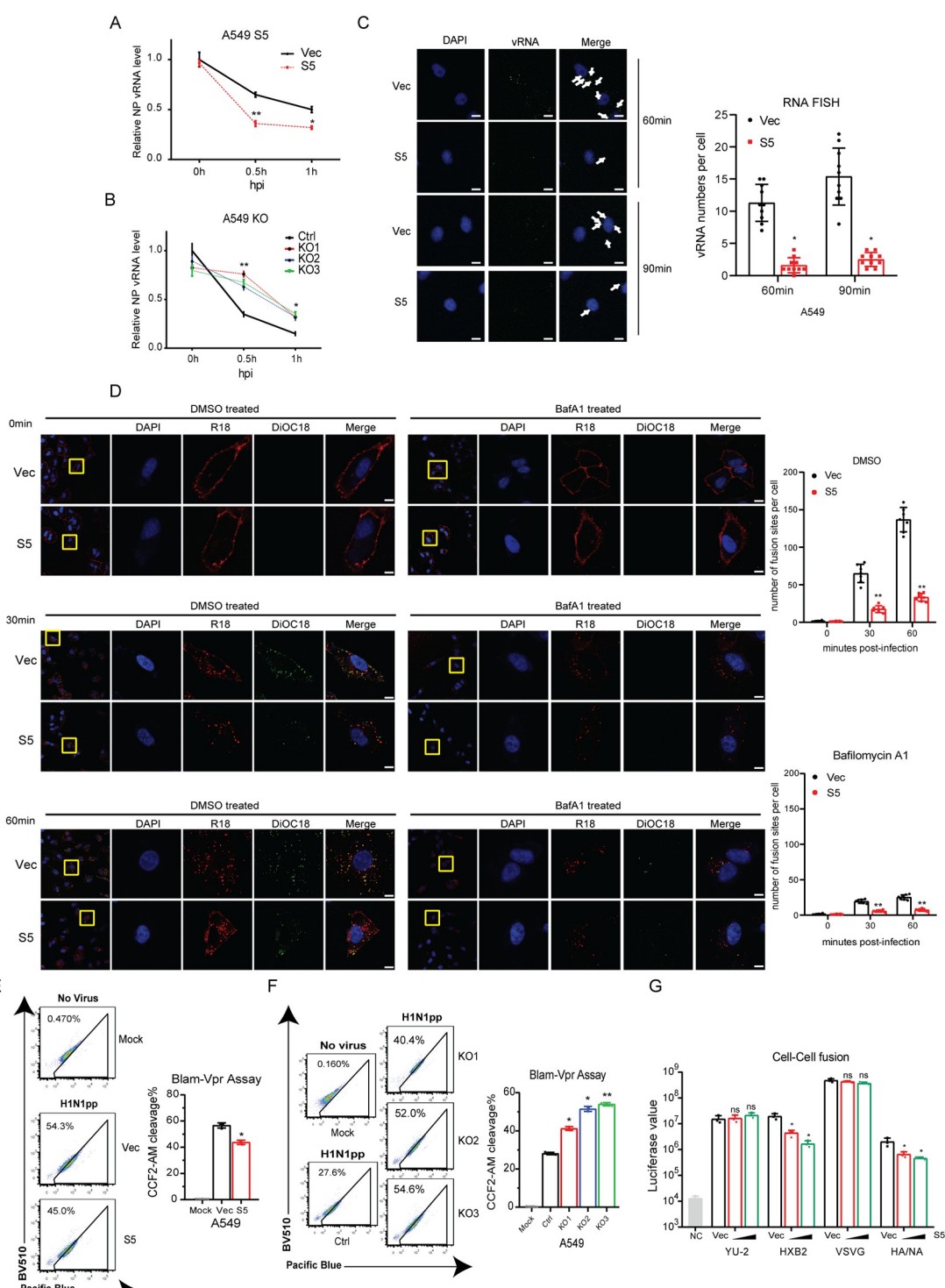

**Fig 3. SERINC5 inhibits the fusion of IAV with cellular membrane.** A-B: Levels of IAV NP vRNA in infected A549 SERINC5-overexpressing cells (A) or SERINC5-knockout cells (B). C: A549 cells with SERINC5 overexpression were incubated with IAV WSN (MOI = 50) for 60 min, or 90 min, then fixed and stained for nuclei (blue), PB1 vRNA (green). Scale bar, 10 μm. Ten views were selected and vRNA numbers per cell are summarized in the graph. D: A549 SERINC5-overexpressing cells were infected with DiOC18 and R18 labled WSN (MOI = 20), cells were fixed and stained for nuclei (blue). Virus were shown in red and virus fusion were shown in green. Scale bar, 5 μm. Seven views were selected and number of fusion sites per cell are summarized in the graph. E-F: BlaM assay to measure IAV fusion with target cells. A549 cells with SERINC5 overexpression (20ng p24 antigen) (E) or SERINC5 knockout (10ng p24 antigen) (F) were incubated with IAV WSNpp that contain BlaM-Vpr

for 2 h. Cells with cleaved CCF2-AM were scored with flow cytometry. Percentages of cells with CCF2-AM cleavage are summarized in the graph. G: 293T cells expressing envelope and HIV-1 Tat-flag were co-cultured with TZM-bl cells transfected with SERINC5. HA/NA cell fusion was triggered with acidic medium. After 40 hours, cells were lysed to measure luciferase activity. Results shown are the averages of three independent experiments. NC: Negative control, 293T cells transfect with empty vector and incubated with TZM-bl cells. Statistical significance was calculated by unpaired $t$-test. ns: not significant; *: $P<0.05$; **: $P<0.01$.

similar to that of the red fluorescence by both self-quenching of DiOC18 and fluorescent resonance energy transfer from DiOC18 to R18, whereas the red fluorescence from R18 is partly self-quenched [26]. Influenza virus fusion events can be illustrated by a shift in the fluorescence color of virus particles. The results showed that the number of fusion sites were reduced in SERINC5-overexpressing cells (Fig 3D). Bafilomycin A1 treatment reduced virus fusion, but didn't affect the inhibition by SERINC5. Virus membrane fusion was also examined using BlaM-Vpr-bearing HIV-1 particles. Fusion of virus with cellular membrane is quantified by measuring the enzymatic activity BlaM that is released into the cytoplasm. The results showed that BlaM levels were reduced in SERINC5-overexpressing cells and increased in SERINC5-knockout cells (Fig 3E and 3F). The inhibitory effect of SERINC5 on IAV HA-mediated membrane fusion was further examined in cell-cell fusion assay. 293T cells were co-transfected with envelope and HIV-1 Tat DNA, then co-cultured with TZM-bl cells that were transfected with SERINC5 DNA. TPCK-treated trypsin was used to cleave and activate HA protein, and membrane fusion was triggered with acidic medium. Levels of luciferase reflected the extent of HA-mediated cell-cell fusion, and a strong inhibition by SERINC5 expression was observed (Fig 3G). Together, these data demonstrate that SERINC5 in the target cells inhibits IAV HA-mediated membrane fusion.

It is not immediately clear how SERINC5, which is located at the plasma membrane, inhibits IAV fusion which occurs in the late endosomes. We propose one possibility is that SERINC5 may respond to IAV infection by relocating to endosomes. Indeed, results of confocal microscopy showed that, IAV HA protein was co-localized with endosome and lysosome markers two hours after infection Importantly, SERINC5 protein became co-localized with HA, late endosome and lysosome markers, in contrast to its localization to the plasma membrane in uninfected cells (S7A Fig). We also examined the kinetics of SERINC5 translocation during virus infection. Results showed that SERINC5 was located at the plasma membrane at 0 hour post infection. As infection proceeded, SERINC5 translocated sequentially to early endosome (EEA1), late endosome (Rab7) and lysosome (LAMP1) (S7B Fig). In support of this result, we observed co-localization of HA-bearing retrovirus particles with SERINC5, but not the VSV-G-bearing retrovirus particles (S7C Fig). These results suggest that upon IAV infection, SERINC5 translocates from the plasma membrane to late endosomes and lysosomes to inhibit IAV entry.

## SERINC5 inhibits infection of multiple IAV strains

IAV is classified into a series of subtypes based on HA and NA. In order to investigate the sensitivity of different IAV subtypes to SERINC5 inhibition, we used HA to pseudotype lentiviral vector and then infected A549 cells that either overexpress SERINC5 or have SERINC5 knocked out. We were able to test three H1 proteins, and each of H2 to H16. The results showed variable sensitivity of HA subtypes to SERINC5 inhibition in the target cells. Among the three HA1 tested, H1 (A/WSN/33) and H1 (A/PR8/34) were inhibited, whereas H1 (A/Brisbane/59/2007) was resistant. From H2 to H16, H4(A/pintailduck/NY/155/1982(H4N3)), H5(A/duck/Egypt/09224F-NLQP/2009(H5N1)), H6(A/chicken/Taiwan/0204/05(H6N1)), H7 (A/Shanghai/4664T/2013(H7N9)), H8(A/duck/Yangzhou/02/2005(H8N4)) and H9(A/Korea/

KBNP-0028/2000(H9N2)) were strongly inhibited by SERINC5, the other HA subtypes were not affected by SERINC5, as shown by the decrease of luciferase activity in SERINC5-overexpressing A549 cells (Fig 4A) or the increase in A549 knockout cells (Fig 4B). VSV-G protein is resistant to SERINC5 restriction, and was included in the experiments as a control. When SERINC5 was overexpressed in virus producer cells, the same HA subtypes were inhibited when used to pseudotype the lentiviral particles (Fig 4C and 4D), and this inhibition was antagonized by HIV-1 Nef protein (Fig 4E and 4F). We then measured the effect of SERINC5 on the infection of IAV strains WSN (A/WSN/33), PR8 (A/PR/8/34), P09 (A/H1N1pdm09) and H3N2 (A/Texas/50/2012) in the SERINC5-overexpressing A549 cells. WSN, PR8 and P09 viruses were inhibited by SERINC5, whereas the H3N2 virus was resistant (Fig 4G). Together, these data demonstrate that SERINC5 selectively inhibits multiple IAV strains.

## HA glycosylation modulates IAV sensitivity to SERINC5 restriction

Next, we investigated the differential sensitivity of IAV HA proteins to SERINC5 inhibition. HA undergoes N-linked glycosylation at several sites [27]. N-linked glycosylation of the globular head of HA is known to modulate the antigenicity, fusion activity, virulence, receptor-binding specificity, and immune evasion of IAV [28]. It had been reported that the Russian-flu H1N1 strain has more glycosylation sites on the head of HA than the 1918 and 2009 pandemic viruses[29], also the number of N-glycosylation site in HA head has increased from 2 to 6 or 7 in 2013 H3N2 [30,31]. We thus propose that HA N-linked glycosylation may modulate IAV sensitivity to SERINC5 restriction. In order to test this possibility, we first scored the N-glycosylation sites in HA1 without signal peptide of H1 to H16 using the NetNGlyc server 1.0 [32,33] (S1 Table). The data showed that SERIC5-sensitive HAs tend to have one or fewer N-glycosylation sites in the HA head domain (73 to 222), whereas SERINC5-resistant HAs have more than two N-glycosylation sites (S8A Fig). In addition, we also observed that 8 out of the 10 SERINC5-resistant HAs have one N-glycosylation site at the residues 160/162, which has been reported to modulate viral entry and neutralization resistance [34], whereas 7 of the 8 SERINC5-sensitive HAs do not have this N-glycosylation site (S8B Fig). Together, these data suggest a correlation between the number as well as the positions of HA glycosylation sites and HA sensitivity to SERINC5 restriction.

Given the profound sequence divergence of different HA subtypes which hampers the effort of identifying the key amino acids or motifs that determine the sensitivity to SERINC5, we took advantage of our observation that among the three HA1 proteins, WSN and PR8 HA were sensitive to SERINC5 as opposed to the Brisbane HA which displayed resistance. When we aligned the sequences of HA head domain of the WSN, PR8 and Brisbane strains (Fig 5A), we found that one of the major differences was the N-glycosylation site. The Brisbane HA has the N-glycosylation motifs (N-X-S/T, where X is not P) at position 87, 125 and 160, whereas WSN HA has one such site at position 125 and PR8 HA has none of these sites.

To investigate whether these N-glycosylation sites affect HA sensitivity to SERINC5, we introduced the glycosylation site to WSN and PR8 HA at position 87, 127, 155 and 160 either individually or at all four sites (Fig 5B), as reported [35]. Results showed that, when all four sites were mutated to be suitable for N-glycosylation, such mutated WSN (WSN-4N) or PR8 (PR8-4N) became resistant to SERINC5 inhibition when used to pseudotype virus particles (Fig 5C and 5D). When we tested HA mutants with individual site changed to N-glycosylation motif, resistance to SERINC5 inhibition was observed for mutations at position 127, 155 and 160, not for 87 (Fig 5C and 5D). It is interesting to note that position 127, 155 and 160 are located to the center top of HA head whereas position 87 is located to the edge, which suggests the importance of glycosylation at the center top of HA head in conferring resistance to SERINC5.

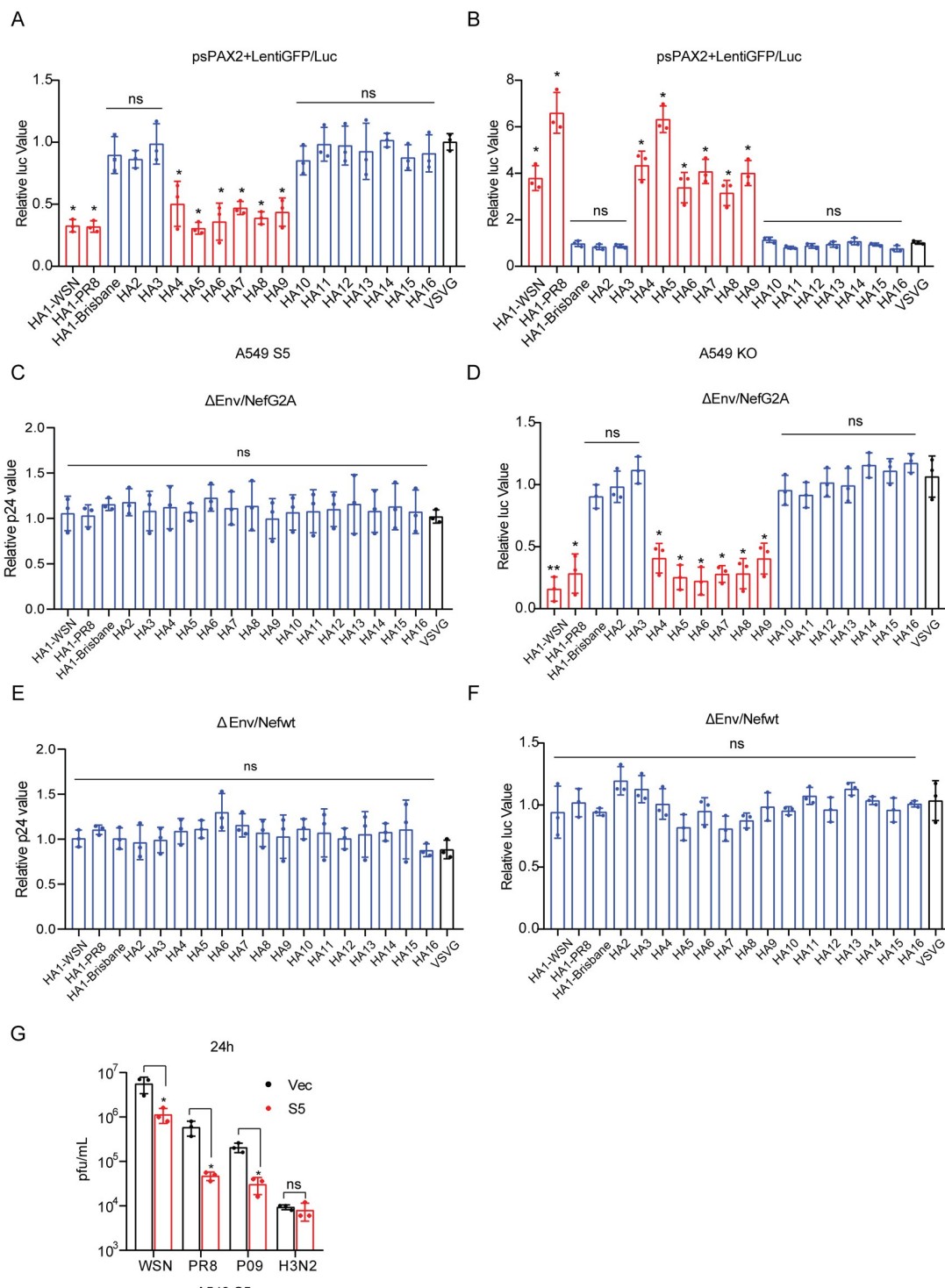

**Fig 4. SERINC5 inhibits multiple IAV strains.** A-B: Pseudoviruses were produced by transfecting HEK293T cells with plasmid DNA psPAX2, LentiGFP/Luc, pCAGGS-HA (expressing H1 to H16), pCAGGS-NA(WSN), or VSV G. The pseudoviruses were then used to infect A549 cells that either overexpress SERINC5 (A) or have SERINC5 knocked out (B). Luciferase activity was measured to report viral infection. C-F: ΔEnv/NefG2A HIV-1^NL4-3 (C, D) or ΔEnv/NefWT HIV-1^NL4-3 (E, F) was used to produce pseudoviruses carrying IAV HA protein (HA1 to HA16) in the presence of SERINC5. Viral particle production (C, E) and viral infectivity (D, F) were measured, and the results of three independent experiments are shown. G: A549 cells with SERINC5 overexpression were infected with A/WSN/33, A/PR/8, A/H1N1pdm09 or A/H3N2 (MOI = 0.05). Titers of IAV in the supernatants were determined by viral plaque assays. Statistical significance was analyzed by unpaired *t*-test. ns: not significant; *: *P*<0.05; **: *P*<0.01.

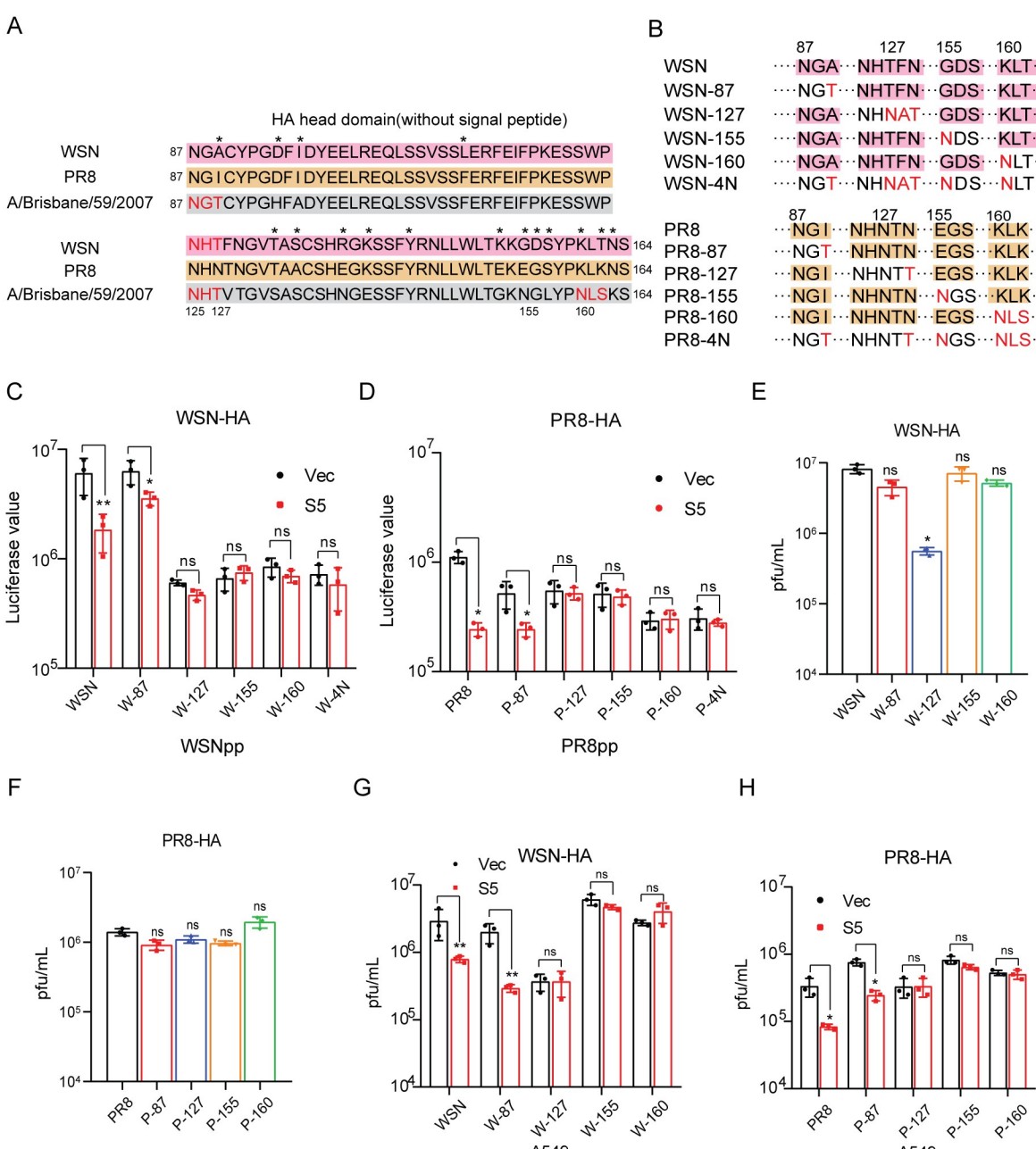

**Fig 5. HA glycosylation sites modulate the sensitivity to SERINC5.** A: The glycosylation sites in the HA head domain of HA1-WSN, HA1-PR8 and HA1-A/Brisbane/59/2007. Different amino acids are marked with asterisks. N-glycosylation site were highlighted with red. B: Schematic representation of IAV WSN and PR8 HA mutations. C-D: HA mutants were used to pseudotype lentivirus particles that were produced by transfecting HEK293T cells with plasmid psPAX2, LentiGFP/Luc and pCAGGS-NA. Viruses of equal viral p24$^{Gag}$ amounts were used to infect A549 cells with SERINC5 overexpression (C) or SERINC5 knockout (D). Luciferase activity was measured to report viral infection. E-H: IAV was produced by transfecting HEK293T cells with the mutated WSN-HA (E, G) or PR8-HA (F, H) and the rest 7 viral DNA segments. Titers of viruses were determined by viral plaque assays (E, F). After infecting A549 cells that stably overexpress SERINC5 at MOI = 0.01 for 24 h, titers of IAV in the supernatant were determined by viral plaque assay (G, H). Results shown are the averages of three independent experiments. Statistical significance was calculated with unpaired $t$-test. ns: not significant; *: $P < 0.05$; **: $P < 0.01$.

Lastly, we examined the effect of the above HA mutations on SERINC5 inhibition of IAV infection. We first prepared IAV by transfecting HEK293T cells with the mutated WSN HA or PR8 HA and the other 7 WSN plasmids using the IAV 8 plasmid reverse genetics system. Among the HA mutations, only mutation at position 127 decreased the production of infectious WSN, no effect was observed for PR8 (Fig 5E and 5F). When all four positions were mutated in HA, no infectious IAV was produced. We then used the wild type or mutated IAV to infect SERINC5-expressing A549 cell line, and observed that SERINC5 reduced the infection of wild type IAV and IAV mutant with N87 HA, but exerted no inhibition for HA mutants 127, 155 and 160 (Fig 5G and 5H). We further investigated whether these glycosylation mutants are restricted by SERINC5 at the membrane fusion step using DiOC18 and R18 dually labeled WSN viruses, BlaM-vpr assay, and cell-cell fusion model (S8C–S8E Fig). Results of these three assays consistently showed that restoring glycosylation site at position 127, 155 and 160, but not 125, created resistance to the inhibition of HA-mediated membrane fusion by SERINC5. We have thus examined the glycosylation of WT HA and its mutants in 6% PAGE, and these HA mutants were properly glycosylated as shown by the results of Western blots examining PNGase-treated sample (S8F Fig). The 87, 127, 155 and 160 HA mutants, which gained one new glycosylation site, showed apparent band shift to higher molecular weights, and a much larger band shift was seen for the 4N mutants which have four new glycosylation sites. After PNGase treatment, the WT HA and all mutants were detected at the same band position in the Western blots, demonstrating that the band shift of the HA mutants was caused by glycosylation. Together, these data support a role of HA head domain glycosylation in countering SERINC5 inhibition.

## Discussion

The antiviral function of SERINC5 was first reported for inhibiting HIV-1. SERINC5 acts by being incorporated into HIV-1 particles and impairing envelope protein-mediated viral entry. Murine leukemia virus (MLV) and equine infectious anemia virus (EIAV) were subsequently shown also subject to SERINC5 inhibition. It remains undetermined whether the antiviral activity of SEERINC5 is limited to inhibiting retroviruses as an antiretroviral protein, or also applies to non-retroviruses. In support of a broader antiviral spectrum for SERINC5, using retrovirus-based pseudoviral system, the entry function of envelope proteins from feline endogenous retrovirus (RD114), IAV (H7), rabies virus and lymphocytic choriomeningitis virus was shown to be restricted by SERINC5[36]. In this study, we further show that SERINC5 inhibits IAV infection, which strongly supports that SERINC5 indeed inhibits viruses beyond retroviruses.

SERINC5 is known to impair the infectivity of HIV-1 particles by becoming a part of the virions. Therefore, SERINC5 exhibits its anti-HIV-1 activity in the virus producer cells, not in the virus target cells. Interestingly, our results showed that SERINC5 inhibits IAV infection when present in either the virus producer cells or the virus target cells. For example, SERINC5 reduces the infectivity of IAV HA pseudotyped lentiviral particles by 10-fold when expressed during virus production (Fig 1A). Similar inhibition was observed when SERINC5 was expressed in the virus target cells (Fig 2A and 2B). The same results were obtained when the single-cycle IAV was tested (Fig 2H and 2I). When expressed in virus target cells, SERINC5 dampens the very early stage of IAV infection (S4 Fig).

It is unclear how SERINC5 in virus target cell prevents IAV infection. IFITM3 is known to deter IAV entry in virus target cells by impairing the fusion of viral membrane with cellular membrane. Our results showed that SERINC5 also impedes this membrane fusion process (Fig 3D–3F), thus inhibits IAV entry into the target cells. Since SERINC5 can modify lipids, it

is possible that SERINC5 changes the property of cellular membrane, which curtails fusion with viral membrane. IFITM3 is localized in late endosomes where IAV completes entry. However, SERINC5 is mostly found at the plasma membrane, thus is not naturally positioned at the site of IAV entry. Strikingly, the imaging data of IAV-infected cells showed that SERINC5 translocated from the plasma membrane to late endosomes and co-localized with IAV particles (S7 Fig), which suggests that along with endocytosis of IAV particles following the interaction of viral HA protein with the sialic acid receptor, the plasma membrane-associated SERINC5 becomes part of the endocytosed vesicle which is destined to become late endosomes.

Although both IFITM3 and SERINC5 inhibit virus entry, they are distinct from each other in some important properties. For example, different from IFITM3, SERINC5 expression is not induced by interferon nor LPS treatment [3]. Yet, SERINC5 has been shown to enhance MDA5-mediated IFN signaling [13], and SERINC5 can interact with MAVS and TRAF6 to inhibit HIV-1 infection [37]. IFITM3 inhibit a much broader range of viruses including IAV, HIV-1, Marburg GP, Ebola GP, SARS-CoV S and many others [38], whereas the antiviral spectrum of SERINC5 is relatively limited to retroviruses in addition to the newly reported inhibition of IAV, LCMV and RV [3]. Their differential antiviral ranges may stem from the specific molecular mechanisms IFITM3 and SERNC5 employ to inhibit virus entry. It will be highly interesting to systematically examine and correlate the sensitivities of different HA subtypes to these two antiviral factors. In the literature, some HA types, such as HA1, HA3, HA5 and HA7 were tested against IFITM3, and were all inhibited [38]. No study has comprehensively tested the full spectrum of HA subtypes against IFITM3 as we have done against SERINC5. The results are expected to identify any role of glycosylation in response to IFITM3 as we observed here with SERINC5, and provide unique insight into the mechanistic details of how IFITM3 and SERINC5 differentially inhibit virus entry.

A wide range of sensitivity toward SERINC5 inhibition has been reported for different viral envelope proteins. For example, VSV glycoprotein and EBOV glycoprotein are completely resistant to SERINC5. Even for envelope proteins of the same virus, such as HIV-1, some isolates (such as HXB2) are strongly inhibited, others (such as YU-2) are refractory to SERINC5 inhibition. We observed similar variations in the response of IAV HA subtypes to SERINC5. Among the 16 HA subtypes we examined, HA1, HA4 to HA9 are more susceptible to SERINC5 restriction than the others using the pseudotyped lentiviral particles (Fig 4A and 4B). We also noted that different isolates of the same subtype may respond differently to SERINC5. For example, the HA1 proteins of WSN and PR8 are sensitive, whereas HA1 of Brisbane is resistant to SERINC5 (Fig 4G). It is unknown whether the resistance to SERINC5 is under pressure selection, or it is simply associated with a certain property or function of HA that drives IAV infection.

The molecular mechanisms are unknown behind the diverse SERINC5 susceptibility of envelope proteins of different viruses or different strains of the same virus. Our results suggest a role of glycosylation in this varying phenotype. First, we observed the correlation between the sensitivity to SERINC5 inhibition and the number of N-glycosylation site in the HA head domain (Fig 5 and S1 Table). More importantly, introducing new glycosylation sites at position 127, 155 and 160 either individually or in combination transformed WSN or PR8 from being sensitive to being resistant to SERINC5 inhibition (Fig 5G and 5H). We have thus examined WT HA and its mutants in 6% PAGE, and these HA mutants were properly glycosylated as shown by the results of Western blots examining PNGase-treated sample (S8F Fig). We also observed slight differences in the migration between the 87, 127, 155, and 160 HA mutants. This might be a result of different carbohydrate chains that might have been added to each of these new sites, in light of the reported high mannose, hybrid type and complex chains

detected in HA1 [39,40]. Based on the prediction by the glycosylation sequons, WSN N125 is a potential glycosylation site. Tandem mass spectrometry analysis of A/New Caledonia/20/1999 showed a vast heterogeneity of glycan compositions at the N125 position, which supports glycosylation of WSN N125 [41]. We also observed that the 127 WSN HA mutant, which change glycosylation positions rather than numbers, shows slight differences in the migration compared with 87 and 155 HA mutants. It needs further studies to explain the band shift of 127 WSN HA mutant. These data suggest that both the number of glycosylation sites and their positions contribute to the sensitivity of HA to SERINC5 restriction.

Glycosylation may modulate HA sensitivity to SERINC5 inhibition by a few possible mechanisms. First, HA with different glycosylation profiles may show different degrees of recognition by SERINC5. SERINC5 has been reported to interact with HIV-1 envelope glycoprotein as one mechanism of antagonizing HIV-1 entry [42]. It is likely that SERINC5 may also directly target HA and this interaction can be modulated by the state of HA glycosylation. Second, viral envelope-mediated cell entry requires extensive conformation changes of envelope protein after engaging the receptor, and SERINC5 may adversely alter envelope protein conformations. Since glycosylation is known to regulate protein folding and conformation [43–45], different glycosylation profiles may allow the viral envelope protein adopt different conformations which exhibit different sensitivities to SERINC5 targeting and inhibition.

It is worth testing whether glycosylation also influences the susceptibility of other viral envelope proteins. For example, HIV-1 Env is heavily glycosylated [46]. Glycosylation can protect Env from binding by neutralizing antibodies, modulate Env conformations, and can even become an epitope of host antibodies. Sequences of Env from different HIV-1 strains are quite divergent, often vary in the number and position of glycosylation sites likely as a result of evading host immunity from both the adaptive and innate arms including host restriction factors like SERINC5. It will be informative to compare the glycosylation profiles of different HIV-1 Env glycoproteins and their sensitivities to SERINC5 inhibition. The findings are expected further strengthen the conclusion of this study on modulation of SEINC5 restriction by envelope glycosylation.

In summary, our data demonstrate SERINC5 restriction of IAV and also reveal the heterogeneity of IAV susceptibility to SERINC5. We further show that SERINC5 inhibits IAV in both virus producer and virus target cells. Intriguingly, IAV HA has the mechanism of escaping SERINC5 restriction by altering glycosylation.

## Materials and methods

### Cells

Human embryonic kidney cells (HEK293T, CRL-2316), human lung adenocarcinoma cells (A549, CCL-185), human cervical cancer cells (HeLa, CCL-2), and Madin-Darby canine kidney cells (MDCK, CRL-2935) were purchased from the American Type Culture Collection (ATCC). TZM-bl cells were obtained from NIH AIDS Reagent Program (catalog number 8129). HeLa CD4+ cells were obtained from NIH AIDS Reagent Program (catalog number 154). A549 stable cell lines overexpressing SERINC5 or empty vector were generated by transfection of A549 with pQCXIP-SERINC5-myc or pQCXIP. After puromycin selection (0.8 μg/ml), cell clones were obtained by dilutions in 96-well plates. All cells were cultured in DMEM with 10% FBS (Hyclone), 1% penicillin (100 U/ml) and streptomycin (100 μg/ml) (Thermo Fisher Scientific).

### Plasmid and reagents

The ΔEnv/Nefwt plasmid was engineered by replacing two amino acids of Env at positions 39 and 40 into two consecutive termination codons through site-directed mutagenesis. The

ΔEnv/NefG2A mutation was generated by mutating the second amino acid of Nef from G to A based on ΔEnv/Nefwt plasmid [8]. The ΔEnv/NefG2A-GFP plasmid was generated by G to A mutation of Nef based on ΔEnv/Nefwt-GFP [47]. The envelope expressing plasmids YU-2, HXB2, VSVG and WSN HA, NA, M1 and M2 were synthesized and constructed into the *Eco*RI/*Xho*I sites in the pCAGGS vector (BCCM). Human SERINC-expression plasmids pQCXIP-SERINC-myc were constructed by inserting the SERINC1 to SERINC5 cDNA into the *Not*I/*Bam*HI sites with C-Myc in the pQCXIP vector (Clontech, 631514). Lentiviral vector system contains psPAX2 (Addgene, #12260) and LentiGFP/Luc (gift from Dr. Zhaohui Qian [48]). SERINC5 expressing plasmid with mCherry tag was constructed by inserting SERINC5 cDNA into *Eco*RI/*Bam*HI sites of the mCherry-C1 vector (Addgene, #54563). BlaM-Vpr (ARP-11444), LTR-Luc (ARP-4789) and Tat-Flag (ARP-10453) plasmids were obtained from NIH AIDS Reagent Program. The following HA DNA sequences were synthesized and sub-cloned into the *Eco*RI/*Xho*I sites in the pCAGGS vector, A/PR/8/3(H1N1) (Genbank: EF 467821.1), A/Brisbane/59/2007(H1N1) (Genbank: CY 163864.1), A/Singapore/1/1957(H2N2) (Genbank: CY 125894.1), A/Brisbane/10/2007(H3N2) (Genbank: KM 978061.1), A/pintail duck/NewYork/155/1982(H4N3) (Genbank: CY 014937.1), A/duck/Egypt/09224F-NLQP/ 2009(H5N1) (Genbank: GU 002686.1), A/chicken/Taiwan/0204/05(H6N1) (Genbank: DQ 376652.1), A/Shanghai/4664T/2013(H7N9) (Genbank: KC 853228.1), A/duck/Yangzhou/02/ 2005(H8N4) (Genbank: EF 061122.1), A/Koreal/KBNP-0028/2000(H9N2) (Genbank: EF 620900.1), A/shorebird/Delaware Bay/10/2004(H10N7) (Genbank: CY 005930.1), A/mallard/ Alberta/245/2003(H11N9) (Genbank: CY 005924.1), A/pintail/Alberta/49/2003(H12N5) (Genbank: CY 005920.1), A/black-headed gull/Sweden/1/99(Genbank: H13N6) (AY 684887.1), A/mallard/Astrakhan/263/1982(H14N5) (Genbank: CY 014604.1), A/duck/Austra-lia/341/83(H15N8) (Genbank: CY 006009.1), A/black-headed gull/Sweden/2/99(H16N3) (Genbank: AY 684888.1). WSN and PR8 HA glycosylation site mutations were created by Q5 site-directed mutagenesis kit (NEB, E0552S).

Plasmid DNA and siRNAs were transfected into cells using PEI (Sigma) and RNAiMAX (Invitrogen), respectively, according to the manufacturer's instructions. SiRNAs targeting SERINC5 were purchased from Thermo Fisher Scientific (AM16708). Bafilomycin A1 were purchased from MCE (HY-100558).

Antibodies used in this article are as follows: p24 (Sino Biological, 11695-R002), Myc (Sigma-Aldrich, C3956), gp120 (Sino Biological, 11233-R011), VSV-G (Sigma-Aldrich, PA1-30138), NP (Millipore Cat# MAB8251), H1N1 HA (Sino bio. Cat# 11055-M08), GFP (Ther-moFisher Cat# PA1-980A), M1 (ThermoFisher Cat# PA5-32222), EEA1 (Sigma-Aldrich Cat# 07–1820), Rab7 (ThermoFisher Cat# PA5-52369), LAMP1 (ThermoFisher Cat# PA5-95849), actin (Sigma-Aldrich, A1978).

## Generation of SERINC5 knockout cells

SERINC5-knockout A549 cell lines were generated with CRISPR-Cas9. A549 cells were trans-fected with lentiCRISPR v2 (Addgene) control vector or CRISPR-Cas9 carrying guide RNAs that target the second exon of SERINC5 (KO1, 5'-GTACGCCCTCTACTTCATTC-3'; KO2, 5'-CGACGACCAGAATGAAGTAG-3'; KO3, 5'-GCTGAGGGACTGCCGAATC-3'), fol-lowed by selection with puromycin (0.8 μg/ml) (ThermoFisher). After SERINC5-knockout cell clones were obtained, genomic DNA of these clones were extracted. Then a 228-nucleotide fragment encompassing exon2 was amplified by PCR using primers 5'- GCTCAGTGCTGTG CGGGCCA-3' and 5'-ACAACAGCACGCATATCCAG-3'. DNA fragments were ligated into T vector and sequenced.

## Influenza A virus propagation and infection

Influenza A virus A/WSN/33(H1N1) was generated by transfecting HEK293T and MDCK co-cultured cells with the 8 plasmids reverse genetics system [49]. Supernatants were collected and used to infect MDCK with MOI = 0.001 for 36 hours. A/PR/8/1934(H1N1) was kindly provided by Dr. Yuelong Shu (China CDC), and propagated in embryonated chicken eggs. A/H1N1pdm09 and A/Texas/50/2012 (H3N2) viruses were kindly provided by Dr. Dayan Wang (China CDC), and propagated in serum-free medium using MDCK cells.

Infections were carried out in a minimum volume of virus maintenance medium (DMEM with 0.5% FBS, 1 µg/ml TPCK-trypsin, 1% penicillin and streptomycin), followed by aspiration of virus and addition of 10% FBS/DMEM. Supernatant and cells were collected at different times post infection.

## Pseudovirus production and virus particle analysis

HIV-1 and influenza pesudoviruses were produced by transfecting HEK293T with 9 µg ΔEnv/Nefwt or ΔEnv/NefG2A, 3 µg HXB2, YU2, VSVG or 6 µg pCAGGS-HA and 3 µg pCAGGS-NA, together with 0.15 µg or 0.6 µg pQCXIP-S5-Myc in 100 mm dishes. Lentiviral vectors were produced by transfection with 9 µg psPAX2, 3 µg lentiGFP/Luc, 3 µg HXB2, YU2, VSVG or 6 µg pCAGGS-HA and 3 µg pCAGGS-NA, together with 0.15 µg or 0.6 µg pQCXIP-S5-Myc. Forty-eight hours after transfection, the supernatant was collected and passed through a 0.2 µm filter. Viral particle production was quantified by $p24^{Gag}$ ELISA.

To determine viral infectivity, equal amounts of $p24^{Gag}$ antigen of viruses were used to infect HeLa-CD4+ or the HIV-1 luciferase reporter cell line TZM-bl in 24-well plates at a density of $4 \times 10^4$ per well. At 48 h after viral infection, cells were lysed with 1×cell lysis buffer (Promega, E1531), and the luciferase activities were determined using the Luciferase assay system (Promega, E1500).

To detect the incorporation of SERINC5 into viral particles, culture supernatants were pelleted through 20% sucrose by ultracentrifugation in an Optima L-100XP ultracentrifuge (Beckman Coulter) at 35000 rpm for 90 min at 4°C. The pelleted virus particles were examined by Western blotting.

## Plaque assay

0.18 Million MDCK cells were seeded in 12-well plates one day before infection. Viruses were serial diluted (1:10) using virus maintenance medium. MDCK were incubated with 250 µl diluted viruses at room temperature for 1 hour (rotated the plate to prevent drying every 15 minutes). Supernatants were discarded, and cells were washed twice with PBS. Then cells were covered with coating buffer (DMEM with 1% Agarose II, 0.075% BSA, 1µg/ml TPCK-trypsin, 1% penicillin and streptomycin). The plate was incubated upside down in the 37°C incubators after the coatings became solid. 72 hours later, viral plaques were scored and the PFU was calculated.

## Flow cytometry

Cells were grown on glass coverslips before infection. After virus infection, cells were suspended in PBS containing 1% paraformaldehyde and resuspended in 400 µl PBS. All cell samples were analyzed on a BD FACS Canto II using the BD Diva software.

## Immunofluorescence assay and RNA FISH

A549 cells were transfected with mCherry-S5, and infected with WSN (MOI 50) for 2 h. Cells were fixed in 4% paraformaldehyde, and permeabilized with 0.1% Triton X-100 for 10 min.

Then cells were stained with anti-EEA1, anti-Rab7, anti-LAMP1, anti-H1N1 HA for 1 h, followed by incubation with anti-mouse IgG conjugated with Alexa Fluor 488 (Thermo Fisher Scientific, A-11088) and anti-rabbit IgG conjugated with Alexa Fluor 647 (Thermo Fisher Scientific, A-21244). Nuclei were stained by DAPI. Images were acquired by Leica TCS SP5 inverted microscope (Leica Microsystems).

RNA-FISH was performed using RNAScope kit (ACD, 323100) according to manufacturer's instructions. Briefly, cells were seeded on glass cover slips. 24 hours later, cells were infected with WSN (MOI 50). After incubated at 4˚C for 1h, cells were washed with ice cold PBS before incubated at 37˚C for different times. Cells were then fixed with 4% PFA for 30 min at room temperature and washed with PBS. Samples were treated with Protease III 15 min at room temperature. Then cells were hybridized with the influenza PB1 probe for 3 h, followed by incubation with PreAmplifier Mix, Amplifier Mix and Label Probe Mix at 40˚C for 30 min, respectively. The nuclei were stained with DAPI. Confocal microscopy and images were acquired by Leica TCS SP5 inverted microscope (Leica Microsystems).

## Gasussia luciferase activity assay

Supernatants of cells infected with gasussia luciferase reporter IAV were collected and luciferase activity was measured using Coelenterazine-h (Promega, S2011) with a Luminometer (Modulus) following the manufacturer's protocol.

## Silic acid linkage expression assay

A549 cell lines either stably overexpressing SERINC5 or with SERINC5 knockout were grown to 70% confluency, dissociated with trypsin free EDTA buffer for 10 min at 37˚C. Cells were then fixed with 4% PFA at room temperature for 15 min and incubated at 4˚C with FITC-conjugated *Sambucus nigra* lectin (SNA, Vector Labs +FL-1301) to detect (α-2,6) sialic acid linkages and biotinylated *Maackia amurensis* lectin II (MAL, Vectore Labs #B-1265), followed by streptavidin-FITC (Invitrogen SA1001) to detect (α-2,3) sialic acid linkages. All cells were analyzed on a BD FACS Canto II.

## Binding assay

SERINC5-overexpressing or SERINC5-knockout A549 cell lines were seeded on glass cover slips and incubated with WSN (MOI = 50/5) at 4˚C for 1 h. Cells were washed with ice cold PBS, then fixed with cold 4% PFA at 4˚C for 15 min and probed with anti-HA rabbit antibody for 2 h at room temperature, followed by anti-mouse Alexa Flour-555 conjugated antibody (Thermo Fisher Scientific, A-21428) for 1 h with 0.5% PBST washes in between, then analyzed by immunofluorescence.

## IAV vRNA quantification

SERINC5-overexpressing or SERINC5-knockout A549 cell lines were incubated with WSN (MOI = 0.1) at 4˚C for 1 h, then washed with ice cold PBS twice to remove the unbound viruses, followed by incubation with warm virus maintenance medium at 37˚C for 0 h, 0.5 h, or 1 h. Cells were harvested by trypsin digestion. Total cellular RNA was extracted and reverse transcribed using Superscript III kit (Invitrogen, 18080–051) following hot start protocol and using IAV HA specific vRNA primers. Real-time PCR was conducted to quantify IAV vRNA as previously described [50].

## BlaM-Vpr pseudovirus fusion assay

Cells were transfected with the following amounts of plasmids in a 100 mm dish: 9 μg ΔEnv/NefG2A, 3 μg BLAM-Vpr, 6 μg pCAGGS-HA and 3 μg pCAGGS-NA. Transfection was carried out using PEI transfection regent. Forty-eight hours after transfection, the supernatant was collected and passed through a 0.2 μm filter. Viral production was quantified by p24$^{Gag}$ ELISA.

To detect BlaM activity with flow cytometry, A549 cell lines were washed with PBS and loaded with the CCF2-AM substrate (Invitrogen, K1023) for 2 h at 37˚C according to the manufacturer's instruction. Following substrate loading, cells were washed and transferred to FACS tubes for flow cytometry analysis.

## Cell-cell fusion

HEK293T cells transiently co-expressing envelope protein and HIV-1 Tat-flag were co-cultivated with TZM-bl cells transduced SERINC5. HA/NA cell fusion was triggered with acidic medium Membrane fusion was then quantified using luciferase assay based on HIV-1 Tat dependent expression of luciferase.

## Dual fluorescent virus fusion assay

Measurement of viral fusion was performed according to the protocol described in [26]. Briefly, WSN was labeled with two fluorescent dyes, octadecyl rhodamine B chloride (R18 Invitrogen, O246) and 3,3'-dioctadecyl-5,5'-di(4-sulfophenyl) oxacarbocyanine, sodium salt (DiOC18 MCE, HY-D0969), in a ratio of 1:2 with final concentrations of 22 μM for R18 and 46 μM for DiOC18. After incubation for 1 h at room temperature, labeled viruses was filtered through a 0.22 μM filter. SERINC5-overexpressing cell lines were incubated with dually labeled WSN particles (MOI = 20) at 4˚C for 1 h, then washed with ice cold PBS twice to remove the unbound viruses, followed by incubation with warm virus maintenance medium at 37˚C for 0 h, 0.5 h, or 1 h. Cells were fixed in 4% paraformaldehyde, and permeabilized with 0.1% Triton X-100 for 10 min. Nuclei were stained by DAPI. Images were acquired by Leica TCS SP5 inverted microscope (Leica Microsystems).

## IAV disassembly assay

SERINC5-mCherry-transfected A549 cells were seeded on glass cover slips and incubated with WSN (MOI = 50) at 4˚C for 1 h. Cells were washed with ice cold PBS twice, followed by incubation with warm virus maintenance medium at 37˚C for 0.5 h, 1 h, or 1.5 h. Cells were then fixed with 4% PFA at room temperature for 15 min, followed by a 10 min permeabilization with 0.2% Triton X-100. Cells were then probed with anti-M1 rabbit antibody (Thermo Fisher Scientific, PA5-32222) for 2 h at room temperature, and anti-rabbit AlexaFlour-647 conjugated antibody (Invitrogen, A-21244) for 1 h with 0.5% PBST washes in between, then examined for fluorescence.

## HA glycosylation mutation verification

293T cells in a 6-well plate were transfected with 1 μg HA glycosylation mutation plasmids. 24 h later, cells were collected and normalized by BCA. Cell lysates were treated in 1x denaturing buffer (100˚C, 10 min). Then the lysates were incubated with or without PNGase F for 3 hours. Glycosylation mutants were examined in 6% polyacrylamide gel followed by Western blotting.

## Statistical analysis

Statistical analyses were performed with GraphPad Prism 7.0 software (GraphPad). Unless otherwise indicated, graphs display mean ± standard deviation (SD) and represent data from at least three independent experiments. Statistical significance was analyzed using unpaired t-test. Significance is indicated by asterisks: $^*$p < 0.05; $^{**}$p < 0.01; n.s., nonsignificant.

## Supporting information

**S1 Fig. Effects of SERINC5 on pseudovirus infectivity.** A-C: Pseudoviruses were produced using ΔEnv/NefWT HIV-1$^{NL4-3}$ that contains the wild type Nef, bearing Env (HXB2), Env (YU-2), VSV-G, or HA/NA, in the presence of different amounts of SERINC5 plasmid DNA. Viral infectivity (A), viral particle production (B) and protein expression (C) were determined as described in Fig 1. D-E: Pseudoviruses were produced with the ΔEnv/NefG2A HIV-1$^{NL4-3}$ plasmid together with IAV HA, NA, M1, M2 plasmids (pCAGGS-HA, pCAGGS-NA, pCAGGS-M1, pCAGGS-M2), in the presence of increasing amounts of SERINC5 DNA. Viral particle production (A) and viral infectivity (B) were measured as described in Fig 1. Results shown are the averages of three independent experiments. Statistical significance was calculated with the unpaired $t$-test. ns: not significant; $^*$: $P$<0.05; $^{**}$: $P$<0.01.
(TIF)

**S2 Fig. Exogenous SERINC5 inhibits infectivity of HA-pseudotyped lentiviral vector.** A-C: Pseudoviruses were produced with the psPAX2 plasmid that expresses viral Gag and Pol, LentiGFP/Luc that expresses luciferase reporter, pCAGGS-HA and pCAGGS-NA that express IAV HA and NA proteins, in the presence of increasing amounts of the SERINC5 plasmid. Equal amounts of viruses as normalized to the levels of p24$^{Gag}$ were used to infect the Hela-CD4+ cells in a 6-well plate at a density of 3×10$^5$ per well. At 48 h after viral infection, half of the cells were lysed to measure luciferase activity (A). The other half cells were fixed to score FITC-positive cells by flow cytometry (B). (C) Viral production was determined by p24$^{Gag}$ ELISA. Results in this figure are the means from three independent experiments. Statistical significance was analyzed by unpaired $t$-test. ns: not significant; $^*$: $P$<0.05; $^{**}$: $P$<0.01; $^{***}$: $P$<0.001.
(TIF)

**S3 Fig. SERINC5 knockout in A549 cells, related to Fig 1.** SERINC5 was knocked out with CRISPR/Cas9 in A549 cells. Single cell clones were selected with puromycin (0.8 ug/ml). Genomic DNA was extracted from the selected cell clones and amplified for exon 2 of SERINC5 gene. The PCR products were cloned and sequenced.
(TIF)

**S4 Fig. SERINC5 inhibits early steps of IAV infection.** A-B: HEK293T cells were transiently transfected with SERINC5 DNA, then infected with A/WSN/33 (MOI = 0.05). Supernatants were harvested 24 h.p.i. Virus titers in the culture supernatants were determined by plaque assays. Levels of SERINC5, viral NP and actin in the infected cells were determined by Western blotting. C-D: HEK293T cells were transfected with SERINC5 siRNA, then infected with virus A/WSN/33 (MOI = 0.01). Viruses and infected cells were harvested 24 h.p.i. (C) SERINC5 mRNA level was quantified by RT-PCR. (D) Virus titers were determined by plaque assays. E: HEK293T cells were transfected with SERINC cDNA S1 to S5, then infected with WSN (MOI = 0.01) for 16 h. Titers of IAV in the supernatants were determined by viral plaque assays. F: Images of A549 cells that were transiently transfected with mCherry-SERINC5 and infected with WSN (MOI = 1) for 4 hours, stained for nuclei (blue), HA (white) and SERINC5

(red). Scale bar, 20 μm. G: Images of WSN (MOI = 1) infected, SERINC5-overexpressing cell lines at 4 h.p.i., stained for nuclei (blue) and NP (green). Scale bar, 20 μm. Statistical analysis is shown in the columns. Results shown are the averages of three independent experiments. Statistical significance was calculated by unpaired $t$-test. ns: not significant; *: $P<0.05$; **: $P<0.01$.
(TIF)

**S5 Fig. SERINC5 does not affect IAV binding to the cell surface.** A: Levels of (α-2,6) and (α-2,3) sialic acid linkages at the surface of A549 SERINC5-overexpressing cells Cells were fixed and incubated with FITC-conjugated *Sambucus nigra* lectin (SNA) to detect (α-2,6) sialic acid linkages and biotinylated *Maackia amurensis* lectin II (MAL, Vector Labs #B-1265) to detect (α-2,3) sialic acid linkages, followed by streptavidin-FITC (Invitrogen SA1001). Percentages of FITC-positive cells are summarized in the bar graph. B: IAV binding to A549 SERINC5-over-expressing cells. After incubation with WSN (MOI = 50/5) at 4°C for 1 h, cells were fixed and stained for nuclei (blue) and HA (red). Scale bar, 30 μm. C: Levels of (α-2,6) sialic acid linkages at the surface of A549 SERINC5-knockout cells. Cells were treated in figure A. Percentages of FITC-positive cells are summarized in the bar graph. D: IAV binding to A549 SERINC5-k-nockout cells. After incubation with WSN (MOI = 50) at 4°C for 1 h, cells were fixed and stained for nuclei (blue) and HA (red). Scale bar, 30 μm. E-F: The original luciferase value in Fig 4A and 4B. Results shown represent three independent experiments. Statistical significance was calculated by unpaired $t$-test. ns: not significant; *: $P<0.05$.
(TIF)

**S6 Fig. Effect of SERINC5 on the early steps of IAV infection.** A: A549 SERINC5-knockout cells were incubated with IAV WSN at MOI = 20 for 60 min, or 90 min, then fixed and stained for nuclei (blue), PB1 vRNA (green). Scale bar, 10 μm. Ten views were selected and vRNA numbers per cell are summarized in the graph. B: IAV disassembly was examined by staining for M1 in A549 cells that were transiently transfected with mCherry-S5. IAV WSN infection was performed with MOI = 50 for 30 min, 60 min, or 90 min. Cells were fixed and stained for nuclei (blue), M1 (white) and S5 (red). Major difference in M1 staining were pointed out with white arrow. Scale bar, 10 μm. Five views were selected and M1 staining in mCherry positive cells are summarized in the graph.
(TIF)

**S7 Fig. SERINC5 translocate to endosome/lysosome during IAV infection.** A: A549 cells were transiently transfected with mCherry-S5, then incubated with (MOI = 50) or without WSN for 2 h. Cells were fixed and stained for nuclei (blue), EEA1, Rab7, LAMP1 (green), S5 (red), HA (white). Scale bar, 10 μm. B: A549 cells were transiently transfected with mCherry-S5, then incubated with WSN at MOI = 50 for 0 h, 1h and 2h. Cells were fixed and stained for nuclei (blue), EEA1, Rab7, LAMP1 (green), S5 (red), HA (white). Scale bar, 10 μm. C: A549 cells were transiently transfected with mCherry-S5, then incubated with HA/NA or VSV-G pseudo particles (50ng p24 antigen) for 2h. Cells were fixed and stained for nuclei (blue), LAMP1 (green), S5 (red), HA/VSV-G (white). Scale bar, 10 μm.
(TIF)

**S8 Fig. HA glycosylation modulates sensitivity to SERINC5 restriction.** A: Statistical analysis of the N-glycosylation site numbers in HA head domain (90 to 240). B: Statistical analysis of the single N-glycosylation site in 160/162 position. C: A549 SERINC5-overexpressing cells were infected with DiOC18 and R18 labled WSN and WSN HA mutation viruses (MOI = 20), cells were fixed and stained for nuclei (blue). Virus were shown in red and virus fusion were shown in green. Scale bar, 5 μm. Seven views were selected and number of fusion sites per cell are summarized in the graph. D: BlaM assay to measure IAV fusion with target cells. A549

cells with SERINC5 overexpression (20ng p24 antigen) were incubated with IAV WSNpp that contain BlaM-Vpr for 2 h. Cells with cleaved CCF2-AM were scored with flow cytometry. Percentages of cells with CCF2-AM cleavage are summarized in the graph. E: Hela cells expressing WSN HAs and HIV-1 Tat-flag were co-cultured with 293T cells transfected with SERINC5 and LTR-Luc. Cell fusion was triggered with acidic medium. After 40 hours, cells were lysed to measure luciferase activity. NC: Negative control, 293T cells transfect with empty vector and incubated with TZM-bl cells. F: Western blot verify the glycosylation of HA mutation. 293T cells transfected with 1ug HA protein expression plasmids. 24h later cells were collected and treated with or without PNGase. Samples were resolved on 6% PAGE. Results shown are the averages of three independent experiments. Statistical significance was calculated by unpaired $t$-test. ns: not significant; *: $P<0.05$; **: $P<0.01$.
(JPG)

**S1 Table. The glycosylation site on HA in different strains.**
(TIF)

## Author Contributions

**Conceptualization:** Fei Zhao, Fengwen Xu, Xiaoman Liu, Fei Guo.

**Data curation:** Fei Zhao, Fengwen Xu, Xiaoman Liu, Yamei Hu.

**Formal analysis:** Xiaoman Liu, Yamei Hu, Liang Wei, Zhangling Fan, Liming Wang, Yu Huang, Shan Mei.

**Funding acquisition:** Chen Liang, Fei Guo.

**Resources:** Li Guo, Long Yang, Shan Cen.

**Supervision:** Shan Mei.

**Validation:** Fei Zhao, Fengwen Xu, Xiaoman Liu.

**Writing – original draft:** Fei Zhao, Fengwen Xu.

**Writing – review & editing:** Jianwei Wang, Chen Liang, Fei Guo.

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
