## [Decision Letter · Decision Letter 0]

17 Jan 2022

Dear Dr. Guo,

Thank you very much for submitting your manuscript "SERINC5 restricts influenza virus infectivity" for consideration at PLOS Pathogens. As with all papers reviewed by the journal, your manuscript was reviewed by members of the editorial board and by several independent reviewers. In light of the reviews (below this email), we would like to invite the resubmission of a significantly-revised version that takes into account the reviewers' comments.

The authors must perform additional experiments using authentic influenza viruses (see reviewer 1's comment).

In addition, the authors need to perform experiments that enhance the mechanistic understanding of their findings (see comments from reviewers 1 and 3)

We cannot make any decision about publication until we have seen the revised manuscript and your response to the reviewers' comments. Your revised manuscript is also likely to be sent to reviewers for further evaluation.

Sincerely,

Yoshihiro Kawaoka

Associate Editor

PLOS Pathogens

Alexandra Trkola

Section Editor

PLOS Pathogens

Kasturi Haldar

Editor-in-Chief

PLOS Pathogens

orcid.org/0000-0001-5065-158X

Michael Malim

Editor-in-Chief

PLOS Pathogens

orcid.org/0000-0002-7699-2064

The authors must perform additional experiments using authentic influenza viruses (see reviewer 1's comment).

In addition, the authors need to perform experiments that enhance the mechanistic understanding of their findings (see comments from reviewers 1 and 3)

Reviewer's Responses to Questions

**Part I - Summary**

Reviewer #1: The manuscripts described the roles of SERINC5, a host protein that was previously demonstrated to have anti-retrovirus activity, in influenza A virus (IAV) infection. It has been previously reported that SERINC5 inhibited virus entry of a virus harboring HA from H7 IAV (Diehl et al., 2021). Accordingly, the main findings of the study that authors argued seemed to be the following three:

- SERINC5 inhibited actual infection of AIV.

- SERINC5 inhibited virus membrane fusion steps.

- The glycosylation of HA proteins regulated the inhibition.

On the other hand, although the pseudotyped HIV-based experiments support the second argument well, the major concerns were still lying on the experiments using authentic AIVs. In addition, the mechanisms of how the glycosylation of the HA modulates SERIN5C-mediated AIV inhibition were not demonstrated nor even discussed. Accordingly, the scientific significance of the study was unclear.

Reviewer #2: In their manuscript, Fei Zhao et al. show that SERINC5 can inhibit entry of influenza viruses. Interestingly, this inhibition can take place when SERINC5 is expressed in target or producer cells. The authors also find that different types of influenza viruses differ in their sensitivity to SERINC5 and that low glycosylation levels correlate with sensitivity. Overall, their findings are supported well by the data presented and the findings are of relevance. However, several points need to be addressed.

Reviewer #3: Zhao and colleagues investigated the impact of SERINC5 on influenza A virus (IAV) infection. In brief, they report that expression of SERINC5 in (retroviral) particle producer cells and in target cells reduced particle infectivity and viral entry, respectively. Further, SERINC5 is shown to inhibit authentic IAV infection and evidence is reported that an early step, membrane fusion, is blocked. Finally, it is demonstrated that hemagglutinins (HA) from different IAV subtypes exhibit differential SERINC5 sensitivity and that the presence of certain sequons in HA can confer SERINC5 resistance.

The study reports, for the first time, inhibition of IAV infection by SERINC5. It contains a good amount of largely well controlled data and is of interest to the field. On the downside, it provides little mechanistic insights. The following points should be addressed.

**Part II – Major Issues: Key Experiments Required for Acceptance**

Reviewer #1: BlaM-Vpr pseudovirus fusion assay demonstrated that virus-mediated membrane fusion was inhibited in pseudotyped HIV. On the other hand, the redistribution of M1 was not fully convincing to support the Authors’ arguments. The approach is not a widely accepted method for visualizing the AIV fusion process. The authors should validate the method with appropriate control studies or provide appropriate references for the method. Alternatively, the authors should use more widely accepted methods to evaluate the AIV fusion process.

The authors argued that the number of the N-linked glycosylation sites is related to the susceptibility toward SERINC5-mediated inhibition. However, this was not supported by the following experiment. In WSN-144, the original N-glycosylation sequon at N142 was abrogated, and alternatively, the virus acquired N-glycosylation sequon at N144. Accordingly, the total numbers of the putative N-glycosylation were the same in WT-WSN and WSN-144. Nevertheless, WSN-144 was resistant to SERINC5 inhibition. How do the authors discuss this discrepancy?

Although the authors argued that the glycosylation of HA proteins regulated the SERINC5-mediated inhibition, the underlying mechanisms were totally unknown. Membrane fusions are mediated by HA2, not HA1, subunit of the HA. How does the glycosylation of the globular head domain of the HA regulate the interaction with host factors that interfere membrane fusion process? The authors should investigate whether the membrane fusion process was interfered in glycosylation mutants and discuss the potential mechanisms underlying.

The reviewer wondered about some discrepancies in the control studies. For example, in Fig 1D, virus growth reached 10^7 pfu/mL in control cells (empty vector-transfected cells). On the other hand, in Fig 1E, that reached only 10^5 in control KO cells, although the experimental conditions of both experiments did not differ so much. Similar tendencies were also observed in Fig2 H and I, and Blam-Vpr Assay in Fig 3E and F. Authors should explain why these discrepancies occurred?

In some figures, the scaling is inappropriate. For example, the scaling of Fig 1A ranged in 10^4 to 10^9. Those ranged in 10^4 to 10^6 in Fig 2H, whereas in 10^5 to 10^7 in Fig 2I. These scales inappropriately exaggerate the difference between the two groups, thereby would mislead the readers.

The authors evaluate the expression of sialosides using SNA staining to ensure that the expression of SERINC5 did not affect the receptor expression. Laboratory strains of IAV such as WSN or PR8 often showed altered receptor binding specificity from original human isolates. It should be doubtful whether WSN utilized only 2-6 sialosides for the infection. The authors should evaluate receptor binding preference of WSN, which was used in the present study, or alternatively evaluate the distribution of 2-3 sialosides.

The authors should check whether each of the glycosylation variants properly acquired glycosylation, not potential glycosylation sites, by using PNGase-digestion and western blotting.

Reviewer #2: Major points to be addressed:

- Fig. 1: SERINC5 seems to be incorporated in all types of VLPS except the IAV-VLPs. However, in these samples also cellular levels of SERINC5 are lower. This should be examined more closely. Does the expression of HA/NA lead to reduced SERINC5 levels or is this a transfection issue in this particular experiment?

- Supp. Fig. 5: The assays shown here need a control to show that the assays would pick up a decrease. This is particularly important for quantifying virus binding by immunofluorescence staining of HA. If the input of virus is halved would that be detected in this assay?

- Fig. 3 C-D only show very few cells as examples. These data should be supported by a quantitative analysis on many cells.

- Fig. 4A: Only relative luciferase levels are shown. Did inhibition correlate with infectivity of the VLPs? Absolute luciferase values should be included as supp. fig..

- The authors show convincingly that the level of HA glycosylation determines SERINC5 sensitivity but this exciting finding is not tested for HIV. Given that the authors already have established the HIV VLP system they should test different variants of env that differ in glycosylation. It will strengthen the manuscript if the authors can show that the level of glycosylation matters also in inhibiting other viruses.

- The discussion section is mostly a repetition of the results but no hypothesis on the mechanism is provided. This needs to be revised and a testable hypothesis for the mechanism of action should be included.

Reviewer #3: Figure 3G (cell-cell fusion) is central to the manuscript and should be better controlled. Please include control viral envelope proteins (for instance HIV-Env and VSV-G), in order to demonstrate specificity, and please include a negative control (empty vector), which allows the reader to judge the assay background.

Supplemental figure 6B is unexpected and the data should be bolstered by additional analyses. Thus, please examine the kinetics of this effect and please determine whether this effect is also seen with retroviral particles bearing HA but not VSV-G.

Little information is available on the expression of SERINC5 in relevant IAV target cells. Is SERINC5 expressed in the respiratory epithelium? Is expression upregulated by IFN and if so, does SERINC5 knock-down render IAV infection less susceptible to inhibition by IFN?

It should be discussed whether also for HIV Env the number of sequons correlates with SERINC5 sensitivity. Furthermore, it should be discussed whether IFITM3 sensitivity of HAs (see for instance PMID: 21253575) from different IAV subtypes correlates with SERINC5 sensitivity.

**Part III – Minor Issues: Editorial and Data Presentation Modifications**

Reviewer #1: L244 based on the antigenicity of the HA and NA.

L243 not seasonal H1N1 IAV, but Russian-flu H1N1 strain.

L324 MLV and EIAV, spell out.

Reviewer #2: Minor points to be addressed:

- L. 114 „IAV membrane also carries the M1 and M2 proteins” This statement suggests that M1 is embedded in the IAV envelope but it functions as a matrix protein and is not “carried” by the IAV membrane. This should be reworded.

- L. 178-182 A reference should be given for WSN-Gluc.

- Fig. 2 H-I: In my opinion, “defective virus” is not a suitable heading for these panels. I would say WSN-Gluc.

- L. 275 In my opinion it is confusing to introduce the abbreviation NGS for N-glycosylation site as NGS usually means next generation sequencing. I would spell out N-glycosylation site.

Reviewer #3: Why are SERINC5 levels low upon coexpression of WSN HA/NA (figure 1C)

“IAV membrane also carries the M1 and M2 proteins.” M1 is not an integral membrane protein.

Ideally, figures 5B-D should include controls like neuraminidase treatment. At least, the absence of such controls should be stated.

PLOS authors have the option to publish the peer review history of their article (what does this mean?). If published, this will include your full peer review and any attached files.

Reviewer #1: No

Reviewer #2: No

Reviewer #3: No
---

## [Decision Letter · Decision Letter 1]

12 Apr 2022

Dear Dr. Guo,

Thank you very much for submitting your manuscript "SERINC5 restricts influenza virus infectivity" for consideration at PLOS Pathogens. As with all papers reviewed by the journal, your manuscript was reviewed by members of the editorial board and by several independent reviewers. In light of the reviews (below this email), we would like to invite the resubmission of a significantly-revised version that takes into account the reviewers' comments.

We cannot make any decision about publication until we have seen the revised manuscript and your response to the reviewers' comments. Your revised manuscript is also likely to be sent to reviewers for further evaluation.

Sincerely,

Yoshihiro Kawaoka

Associate Editor

PLOS Pathogens

Alexandra Trkola

Section Editor

PLOS Pathogens

Kasturi Haldar

Editor-in-Chief

PLOS Pathogens

orcid.org/0000-0001-5065-158X

Michael Malim

Editor-in-Chief

PLOS Pathogens

orcid.org/0000-0002-7699-2064

Reviewer's Responses to Questions

**Part I - Summary**

Reviewer #1: The revised manuscript addressed most of the points suggested by reviewers. However, the authors totally misunderstood the interpretation of the PNGase F treatment. The experiment is strikingly important to support the author's argument that glycosylation regulated the sensitivity of the HA to SREINC5. Thereby, the reviewer requires a careful explanation to this issue from the authors.

Reviewer #2: In my opinion, the findings of this revised manuscript are of broad interest to the field and the conclusions are well supported by the data presented.

**Part II – Major Issues: Key Experiments Required for Acceptance**

Reviewer #1: In Fig S8, the authors concluded that the mutant HAs are appropriately glycosylated. On the other hand, there seems no apparent band shifts in mutant HAs (except W-144) compared with WT-HAs. Especially for W-4N and P4N, addition of 4 glycosylation sites should result in significant sift of bands in the WB analysis. This should indicate the introduction of the mutation did not affect the glycosylation profile of the HA. In this situation, one of the main results that that glycosylation regulated the sensitivity of the HA to SREINC5 was not convincing. The reviewer requires a careful explanation to this issue.

Legends for the Fig S8F (L1046¬–1050) are inappropriate. There should not be “average of three independent experiments” nor “statistical significance”.

Reviewer #2: My concerns have been addressed adequately.

**Part III – Minor Issues: Editorial and Data Presentation Modifications**

Reviewer #1: Throughout the manuscript, the reviewer strongly encourages the authors to use the so-called “H3 numbering (based on Wilson et al., 1981)” system for the numbering of the HA protein,

https://www.nature.com/articles/289366a0

In this system, the amino acid corresponding to the matured (without signal peptide) HA of A/Hong Kong/68 (H3N2) will be used for the counting of the position.

L71–72

It seems that RD114 is the abbreviation for the strain name, H7 should be subtype and RABV and LCMV should be the name of viruses. Use the abbreviations in the same category in these parentheses.

L129 (NA.) should be (NA).

L207

MAL is normally used for the abbreviation for MAA-I lectin (Maackia amurensis leucoagglutinin). Maackia amurensis lectin II should be abbreviate to MAL-II, MAA-II or MAH.

https://pubmed.ncbi.nlm.nih.gov/21863598/

L443–453

Using the word Env here is confusing. Do they mean Env protein of Retrovirus, or simply viral envelope protein? Please specify.

L686–687

Text garbling in the review PDF.

Fig 3D/S8C

Number of fusion site per what?

Reviewer #2: (No Response)

PLOS authors have the option to publish the peer review history of their article (what does this mean?). If published, this will include your full peer review and any attached files.

Reviewer #1: No

Reviewer #2: No
---

## [Decision Letter · Decision Letter 2]

19 May 2022

Dear Dr. Guo,

Thank you very much for submitting your manuscript "SERINC5 restricts influenza virus infectivity" for consideration at PLOS Pathogens. Your revised manuscript was reviewed by members of the editorial board and sent out for external review . In light of the comments of reviewer 1 (below this email), we need to request additional information from you. Reviewer 1 noted a serious concern regarding the validity of the blots shown in Fig S8F. Therefore, the editors of PLoS Pathogens request that you provide an uncropped and unedited raw blot image and an explanation of the apparent duplication and manipulation of the blot in the current Fig S8F.

We cannot make any decision about publication until we have seen your response to this matter, the revised manuscript and your response to the remaining comments of reviewer 1. Your revised manuscript will be sent to external review for further evaluation.

Sincerely,

Yoshihiro Kawaoka

Associate Editor

PLOS Pathogens

Alexandra Trkola

Section Editor

PLOS Pathogens

Kasturi Haldar

Editor-in-Chief

PLOS Pathogens

orcid.org/0000-0001-5065-158X

Michael Malim

Editor-in-Chief

PLOS Pathogens

orcid.org/0000-0002-7699-2064

Reviewer's Responses to Questions

**Part I - Summary**

Reviewer #1: The authors re-conducted the experiment to confirm the HA glycosylation; however, the result was not fully convincing. Furthermore, there is a critical concern on FigS8F that was newly provided. Thus, the reviewer again requires a careful explanation of the result from the authors.

**Part II – Major Issues: Key Experiments Required for Acceptance**

Reviewer #1: Although the authors mentioned that the experiment in Fig S8F, which was done on 12.5 % acrylamide gel, was re-conducted on 6% acrylamide gel, the pannels on "PNGase digestion(+)" seem copy-and-paste from the revision 1 figures. There is no description that the experiments on "PNGase digestion (-)" and "PNGase digestion(+)" were conducted under different conditions. Also, scales for the molecular weight in the R1 figure and R2 figure are labeled differently; in the R1 version, bands for deglycosylated HAs came under the 55KD line, but in the R2 version, they came between the 70KD and 55KD lines.

This might be a simple mistake by the authors; however, the authors should carefully explain this dubious result.

To make the situation clear, the reviewer requires the authors to provide the original (unedited) picture of the western-blotting membranes of this experiment.

From the amino acid sequence, the W-127 mutant lost the glycosylation site on 125 and gained an additional glycosylation site on 127. Accordingly, the total number of the glycosylation sites was unchanged. On the other hand, the clear band shift was observed even in this strain. It seemed unlikely that the difference was due to the difference in glycan structures between WT and W127 because the sift in W-127 was larger than those in W-87 and W-155. If the authors claim that site 125 was not glycosylated, the authors should demonstrate it by the site-specific glycoproteomic analysis or refer the appropriate citation.

**Part III – Minor Issues: Editorial and Data Presentation Modifications**

Reviewer #1: None

PLOS authors have the option to publish the peer review history of their article (what does this mean?). If published, this will include your full peer review and any attached files.

Reviewer #1: No
---

## [Decision Letter · Decision Letter 3]

11 Jul 2022

Dear Dr. Guo,

Thank you very much for submitting your manuscript "SERINC5 restricts influenza virus infectivity" for consideration at PLOS Pathogens. As with all papers reviewed by the journal, your manuscript was reviewed by members of the editorial board and by several independent reviewers. In light of the reviews (below this email), we would like to invite the resubmission of a significantly-revised version that takes into account the reviewers' comments.

The authors should perform the experiment requested by Reviewer 1 because the comparison of the protein with and without PNGase treatment should be made under the same conditions (i.e., a 6% gel in this case).

We cannot make any decision about publication until we have seen the revised manuscript and your response to the reviewers' comments. Your revised manuscript is also likely to be sent to reviewers for further evaluation.

Sincerely,

Yoshihiro Kawaoka

Associate Editor

PLOS Pathogens

Alexandra Trkola

Section Editor

PLOS Pathogens

Kasturi Haldar

Editor-in-Chief

PLOS Pathogens

orcid.org/0000-0001-5065-158X

Michael Malim

Editor-in-Chief

PLOS Pathogens

orcid.org/0000-0002-7699-2064

The authors should perform the experiment requested by Reviewer 1 because the comparison of the protein with and without PNGase treatment should be made under the same conditions (i.e., a 6% gel in this case).

Reviewer's Responses to Questions

**Part I - Summary**

Reviewer #1: Although the authors revised the descriptions on the SFig. 8 (PNGase digestion), the settings for this experiment were not appropriate.

**Part II – Major Issues: Key Experiments Required for Acceptance**

Reviewer #1: The authors resolved PNGase (+) samples with 12.5% PAGE gels, whereas PNGase (-) samples with 6% gels.

Resolving the protein with 12.5% gels underestimates the electrophoresis mobility of proteins; this should be why the authors used 6% gels for resolving PNGase (-) samples.

Accordingly, the reviewer again requires the authors to conduct the western blot analysis of both PNGase (+) and (-) samples on 6% PAGE gels. Also, the pairs of PNGase (+) and (-) samples should be resolved strictly on the same gels.

**Part III – Minor Issues: Editorial and Data Presentation Modifications**

Reviewer #1: None

PLOS authors have the option to publish the peer review history of their article (what does this mean?). If published, this will include your full peer review and any attached files.

Reviewer #1: No
---

## [Decision Letter · Decision Letter 4]

30 Sep 2022

Dear Dr. Guo,

We are pleased to inform you that your manuscript 'SERINC5 restricts influenza virus infectivity' has been provisionally accepted for publication in PLOS Pathogens.

Best regards,

Yoshihiro Kawaoka

Associate Editor

PLOS Pathogens

Alexandra Trkola

Section Editor

PLOS Pathogens

Kasturi Haldar

Editor-in-Chief

PLOS Pathogens

orcid.org/0000-0001-5065-158X

Michael Malim

Editor-in-Chief

PLOS Pathogens

orcid.org/0000-0002-7699-2064

Reviewer Comments (if any, and for reference):

Reviewer's Responses to Questions

**Part I - Summary**

Reviewer #1: The authors appropriately addressed the comment from the reviewer.

**Part II – Major Issues: Key Experiments Required for Acceptance**

Reviewer #1: None

**Part III – Minor Issues: Editorial and Data Presentation Modifications**

Reviewer #1: None

PLOS authors have the option to publish the peer review history of their article (what does this mean?). If published, this will include your full peer review and any attached files.

Reviewer #1: No

---

## [Editor Report · Acceptance letter]

7 Oct 2022

Dear Dr. Guo,

We are delighted to inform you that your manuscript, "SERINC5 restricts influenza virus infectivity," has been formally accepted for publication in PLOS Pathogens.

Best regards,

Kasturi Haldar

Editor-in-Chief

PLOS Pathogens

orcid.org/0000-0001-5065-158X

Michael Malim

Editor-in-Chief

PLOS Pathogens

orcid.org/0000-0002-7699-2064